# From Capillary Electrophoresis to Deep Sequencing: An Improved HIV-1 Drug Resistance Assessment Solution Using In Vitro Diagnostic (IVD) Assays and Software

**DOI:** 10.3390/v15020571

**Published:** 2023-02-19

**Authors:** Sofiane Mohamed, Ronan Boulmé, Chalom Sayada

**Affiliations:** 1ABL Diagnostics, 57140 Woippy, France; 2Advanced Biological Laboratories (ABL), 2550 Luxembourg, Luxembourg

**Keywords:** HIVDR, HIV, whole genome, NGS, capillary electrophoresis, drug resistance, algorithm

## Abstract

Background: Drug-resistance mutations were mostly detected using capillary electrophoresis sequencing, which does not detect minor variants with a frequency below 20%. Next-Generation Sequencing (NGS) can now detect additional mutations which can be useful for HIV-1 drug resistance interpretation. The objective of this study was to evaluate the performances of CE-IVD assays for HIV-1 drug-resistance assessment both for target-specific and whole-genome sequencing, using standardized end-to-end solution platforms. Methods: A total of 301 clinical samples were prepared, extracted, and amplified for the three HIV-1 genomic targets, Protease (PR), Reverse Transcriptase (RT), and Integrase (INT), using the CE-IVD DeepChek^®^ Assays; and then 19 clinical samples, using the CE-IVD DeepChek^®^ HIV Whole Genome Assay, were sequenced on the NGS iSeq100 and MiSeq (Illumina, San Diego, CA, USA). Sequences were compared to those obtained by capillary electrophoresis. Quality control for Molecular Diagnostics (QCMD) samples was added to validate the clinical accuracy of these in vitro diagnostics (IVDs). Nineteen clinical samples were then tested with the same sample collection, handling, and measurement procedure for evaluating the use of NGS for whole-genome HIV-1. Sequencing analyzer outputs were submitted to a downstream CE-IVD standalone software tailored for HIV-1 analysis and interpretation. Results: The limits of range detection were 1000 to 10^6^ cp/mL for the HIV-1 target-specific sequencing. The median coverage per sample for the three amplicons (PR/RT and INT) was 13,237 reads. High analytical reproducibility and repeatability were evidenced by a positive percent agreement of 100%. Duplicated samples in two distinct NGS runs were 100% homologous. NGS detected all the mutations found by capillary electrophoresis and identified additional resistance variants. A perfect accuracy score with the QCMD panel detection of drug-resistance mutations was obtained. Conclusions: This study is the first evaluation of the DeepChek^®^ Assays for targets specific (PR/RT and INT) and whole genome. A cutoff of 3% allowed for a better characterization of the viral population by identifying additional resistance mutations and improving the HIV-1 drug-resistance interpretation. The use of whole-genome sequencing is an additional and complementary tool to detect mutations in newly infected untreated patients and heavily experienced patients, both with higher HIV-1 viral-load profiles, to offer new insight and treatment strategies, especially using the new HIV-1 capsid/maturation inhibitors and to assess the potential clinical impact of mutations in the HIV-1 genome outside of the usual HIV-1 targets (RT/PR and INT).

## 1. Introduction

The majority HIV drug resistance (HIVDR) genotyping utilizes the capillary electrophoresis (Sanger) (CE) sequencing method. This technology has been validated for HIVDR determination; it is generally limited to the detection of nucleotide variants and variant haplotype signatures present at 20% prevalence [1,2]. Several studies have clearly demonstrated that HIVDR variants detection between 1% and 20% could improve treatment outcomes [3,4,5,6,7]. It is therefore important to detect mutations at 20% but also minor variants that occur below 20% frequency, using a Next-Generation Sequencing (NGS) method. This is offered by in vitro diagnostics assays with replicable and repeatable validated performance, using a seamless and robust HIVDR interpretation software. Both sequencing methods, capillary electrophoresis and NGS, ended with HIVDR for general anti-HIV treatments and for the monitoring and the interpretation of the new HIV-1 capsid/maturation inhibitors.

Minority variant detection and HIVDR monitoring using NGS were successfully performed within human immunodeficiency virus type 1 (HIV-1) patients [1,8,9,10]. The new capsid and maturation antiretroviral (ARV) treatments have improved patient prognosis [11,12]. However, virological failure of the new ARV has been reported [13,14,15,16,17]. Several HIVDR interpretation algorithms have been widely used for assessing virological response in retrospective analyses: ANRS [18], Stanford HIVdb [19], and IAS-USA [20]. The algorithms have changed over time and use different rules to predict drug resistance. Thus, the interpretation may differ between these algorithms [21], meaning that there is a need to combine all the available renowned algorithms into a single report with the continuous updates of their algorithm versions to optimize the monitoring and the management of HIV-1-infected patients.

The first aim of this study was to evaluate the NGS and capillary electrophoresis, using CE-IVD protocols designed to amplify and subsequently to detect and to assess HIVDR. The second aim was to compare the target-specific method (reverse-transcriptase, protease, and integrase, using separate amplifications) with the new whole-genome HIV-1 sequencing strategy, still using NGS.

## 2. Materials and Methods

### 2.1. Performance Evaluation

The DeepChek^®^ Assays went for analytical and clinical performance evaluation to measure and to report their ability to amplify relevant targets of the HIV-1 genome in order to conduct further downstream analysis, interpretation, and clinical reporting useful for HIV-1 positive patients’ management (subtype characterization, drug-related-mutations detection, and drug-resistance assessment).

### 2.2. Clinical Samples

According to the intended use of the assays, only clinical samples from HIV-1-diagnosed patients with positive viral loads were selected, except for cross-reactivity testing. In total, 301 clinical samples were retrospectively collected from HIV-1 patients for the main testing CE-NGS, and 19 samples for the whole-genome sequencing. Leftover clinical specimens were used as permitted in the context of non-interventional studies (no additional procedure, no unusual diagnostics, and no monitoring). Laboratory sample request forms informed patients that the leftovers of samples could be used for research purposes. The leftovers were used anonymously, ensuring confidentiality. Two sets of experiments were conducted. In both, the viral RNA was extracted from 1 mL of plasma (previously stored at −80 °C), using the MagNa Pure Nucleic Acid Isolation Kit I (Roche Diagnostics). The RNA was eluted in 50 μL of elution buffer according to the instructions for use. A total of 69 samples (Quality Control for Molecular Diagnostics (QCMD), cross-reactions and clinical (B and non-B subtypes)), and 5 amplification negative controls were prepared for the CE/NGS HIV-1 target-specific sequencing. Three different lots with three distinct operators at different times of the day over 5 days were used. For the second sequencing strategy testing, nineteen samples were tested in parallel with whole-genome HIV-1 genotyping and the HIV-1-targeted genotyping (reverse transcriptase, protease, and integrase).

Both procedures amplified the samples using the Applied Biosystems ProFlex PCR System model 3 × 32-well (REF 4484073) (Thermo Fisher Scientific, Walthem, MA, USA). After the library preparation, all samples were sequenced with the Illumina MiSeq and iSeq100 instruments and analyzed with the DeepChek^®^ Software (REF S-12-023, version 3.30) (ABL, Luxembourg) (Whole Genome HIV and HIV modules).

### 2.3. RNA Amplification

To concentrate the RNA, ultracentrifugation (24,000× *g* during 1 h at 4 °C) and a centrifugal filter (Merck KGaA, Darmstadt, Germany) were used for samples with low viral load (<1000 copies/mL for target-specific amplifications and <25,000 copies/mL for whole-genome amplification). RNA was amplified using two kits (i) for the three HIV-1 genomic targets, using the DeepChek^®^ Assay PR/RT (REF 121A24) and INT (REF 122A24) (CE-IVD) (ABL, Luxembourg) and (ii) the DeepChek^®^ Assay HIV Whole Genome (REF 170A24) (CE-IVD) (ABL, Luxembourg). The study design and the capillary electrophoresis/NGS primers for specific target and whole-genome sequencing are presented in Figure 1 and Figure 2. For the analytical performance, a total of 444 amplifications were performed, and 74 samples’ (RT/PR and INT were pooled) NGS outputs were configured and analyzed in batch by the DeepChek^®^ Software (CE-IVD) (HIV module), using a fixed configuration of parameters (algorithms, threshold, and expert system).

### 2.4. Capillary Electrophoresis Sequencing

Antiretroviral-resistance mutations were genotyped using the DeepChek^®^ Assay PR/RT (REF 125A24) and INT (REF 126A24) Sanger sequencing accessory kits (CE-IVD) (ABL, Luxembourg) according to the manufacturer’s instructions. These assays were combined with enzymes, a buffer, dNTPs, and dyes (ABL DeepChek^®^ Assay Sanger Sequencing Reaction kit (references 123A48 and 123A24, for PR/RT and INTtargets, respectively) for the sequencing technology (Applied Biosystems SeqStudio Genetic Analyzer, REF A35644, Thermo Fisher Scientific, Walthem, MA, USA). The related nucleotide sequences (fasta or ABI files) were analyzed to identify the HIV genotypes and the drug-resistant mutants through ViroScore-HIV (REF S-09-014) and DeepChek^®^ software with the HIV module (S-12-023 (HM)) (CE-IVD) (ABL, Luxembourg).

### 2.5. NGS

The libraries of the HIV-1 RT/PR/INT and whole genome (Table 1) were prepared using the DeepChek^®^ NGS Library preparation V1 (REF 116A) (ABL, Luxembourg) following the manufacturer’s protocol. The appropriate volume of each pool of amplicons was adjusted to have a total quantity of 2 ng of amplicon input per sample. The libraries were qualified on an Agilent Technologies Fragment Analyzer system, using a fragment Analyzer DNA Kit (Agilent Technologies, Boeblingen, Germany), following the manufacturer’s instructions, and quantified with the Qubit 2.0 fluorometer, using the dsDNA HS Assay Kit (Thermo Fisher Scientific, Walthem, MA, USA). The resulting libraries were sequenced using the MiSeq system (Illumina, San Diego, CA, USA), and the run was performed by generating 2 × 251 bp read length data during a 39 h run time for whole-genome sequencing and QCMD samples, using target-specific assays (RT/PR and INT). In parallel, libraries were sequenced using the iSeq100 system (Illumina, San Diego, CA, USA), and the run was performed by generating 2 × 151 bp read length data during a 19 h run time for target-specific assays (RT/PR and INT). HIV libraries were sequenced from both ends (forward and reverse).

### 2.6. Selection of Variant Frequency Threshold

Two thresholds (≥3% and ≥20%) were selected for reporting. The 20% threshold was selected as a reference for comparison with the CE sequencing method [22,23]. Mutations were considered significant at a frequency ≥3% among the total number of reads if they were present in both sequence directions. This threshold was selected based on previous results [22,23,24].

### 2.7. Data Analysis

Sequencing analyzers’ outputs (sequences) were mapped against HIV-1 reference (HXB2) and analyzed using tailored bioinformatics pipelines from ViroScore or DeepChek^®^ software and then interpretated using regularly updated HIV Drug Resistance mutations knowledge databases and algorithms. HIVDR interpretations were assessed for ARV, using the ANRS and Stanford HIVdb algorithms (other algorithms with their latest versions are available, such as Grade, Rega, RenaGeno, RIS, or Geno2pheno) (version 3.30.18; Expert System (v2.3); Drug Resistance Rulers algorithm for HIV (v.11.9)). For in silico analyses, during the performance evaluations (cross-reactivity and inclusivity), bacteria, fungi, and virus sequences were taken from NCBI Genomes representative sequences and from NCBI Nucleotide. The human genome reference (GRCh38) was added to the analysis.

### 2.8. ViroScore Software

ViroScore software (Figure 3) stores and organizes sequencing data for analysis from CE sequencing. By providing an integrated chromatogram viewer/editor, it allows the user to analyze, from end-to-end, its samples in a few minutes, from traces cleaning to drug-resistance interpretation, genotyping, and reporting. Being able to follow a patient through time and compare several results is another core part of ViroScore which gives the possibility to aggregate multiple results for comparison (either between different algorithms or between different samples and/or patient over time). Sequences are aligned against the HIV K03455.1 (HXB2) reference, using clustalW for each region that was defined, and variants are called. Chromatogram editor can be used to correct the raw sequences (forward and reverse). Then all selected HIVDR algorithms are applied to the list of mutations for interpretation, while BLAST (basic local alignment search tool, NCBI) or COMET (Context-Based Modeling for Expeditious Typing, Luxembourg Institute of Health) is used on a per-region basis to determine their most likely subtype. Mutations and subtypes’ results are then aggregated and compiled on a final result page and in PDF reports.

### 2.9. DeepChek^®^ Software

DeepChek^®^ software (Figure 4) with its HIV module is a CE-IVD downstream analysis software which allows an automated sequencing analysis mainly from NGS Illumina sequencing data (fastq). It can also store and analyze other NGS outputs from sequencing analyzers from Thermo Fisher, MGI, and Nanopore. It also integrates additional sources, such as capillary electrophoresis sequencing trace files (AB1). Then it analyzes several key regions, such as RT, PROT, INT, and CAP, and offers both drug-resistance interpretation and clinical reporting using various well-defined algorithms, such as Stanford HIVdb, ANRS, or IAS-USA (and also Grade, Rega Institute, RenaGeno, RIS). Sequences are aligned against the HIV K03455.1 (HXB2) reference, using BWA (main bioinformatics pipeline for Illumina sequencing outputs), and split by regions before using BLAST or COMET for determining the most probable subtype. Then variants are called and filtered using an expert system that cleans out unbalanced mutations between forward and reverse reads, as well as too-low-frequency or covered variations (when dealing with NGS data). Finally, all the user-selected HIVDR algorithms are applied to the list of mutations for interpretation before being reported in the final PDF files. DeepChek^®^ software works the same with outputs from either HIV-1 target-specific or whole-genome sequencing. When it comes to capillary electrophoresis data, DeepChek^®^ will automatically assemble the AB1 files and analyze the resulting contig.

## 3. Results

### 3.1. Analytical Limit of Detection

The analytical limit of detection (LOD) was defined as the lowest concentration at which ≥95% of tested replicates were shown to be presumptive positive for the detection of PR, RT, and IN targets of HIV-1. LOD was 1000 copies/mL for HIV-1 subtype B, as reported in Table 2. One hundred percent of sequencing was achieved, with samples having a viral load at 10^6^ copies/mL. Clinical samples with viral loads below or equal 1000 copies/mL were correctly amplified using an ultracentrifugation step and with the RNA centrifugal filter. For HIV-1 whole-genome sequencing, serial dilutions were made with an HIV-1 reference control (LGC SeraCare, SeraSeq, isolate 93/US/141, Catalog #0740-0006, 1,000,000 copies/mL). The concentration level for the DeepChek^®^ Assay Whole Genome HIV-1 Genotyping with observed rates greater than or equal to 95% (limit of detection) was 25,000 copies/mL HIV-1 RNA with the ProFlex3 × 32-well PCR System.

### 3.2. Analytical Cutoff and Cross Reactivity

One hundred percent of samples with an optimal median coverage at a concentration of 1000 copies/mL (assay cutoff) was obtained (Table 3). The median coverage per sample for the three amplicons (PR/RT and INT) was 13,237 reads. For whole-genome sequencing, a median number of 308,739 reads was determined for the 20 samples, with a median of 93% of the reads mapping to HIV-1. No interference substances were reported, as no cross-reactivity occurred with the HCV- and HBV-spiked clinical samples. Thus, the in silico analytical study showed no amplification of organisms other than HIV-1 (viruses, bacteria, or humans).

### 3.3. Clinical

One hundred percent of clinical reproducibility with the QCMD Panel was obtained (Table 4). One hundred percent agreement was found between the iSeq100 and MiSeq analyzers: no interpretation difference was observed between the iSeq100 and MiSeq. The NGS method was more sensitive than the capillary electrophoresis technology. The mutation K43T on the PR was detected with NGS only for both analyzers, iSeq100 and MiSeq, at 19,63% and 19,69%, respectively. Differences of interpretation between ANRS and Stanford algorithms were observed for the RT mutation V179I and for the PR mutations L10I/V, G16E, K20R, L33I, M36V/I, D60E, L63P, H69K, A71T, and L89M/I. For the 301 processed samples, the clinical sensitivity was 99% or 94% for amplifying and obtaining a sequence of good quality for HIV drug-resistance testing and interpretation using the protease and reverse transcriptase or integrase alone, respectively (Table 5). The remaining 1% to 6% could be use errors, assay limitations, or a combination of both. When both assays were combined, the clinical sensitivity was 99%. In Table 6, the concordance percentage between DeepChek^®^ Assay and similar assays available on the market was reported. The three head-to-head comparisons were performed using the DeepChek^®^ Assay, together with an Illumina MiSeq instrument for downstream processing. The only downstream NGS similar assay was the Sentosa^®^ SQ HIV-1 Genotyping (Vela Diagnostics). The two others were based on capillary electrophoresis sequencing. The DeepChek^®^ assay performed as well as the predicates available on the market, with a high concordance (>90%) to amplify HIV-1 RT, PR, and INT.

Out of the 19 samples of the second evaluation, 16 had results available for both the whole-genome sequencing and the target-specific sequencing using next-generation sequencing. There was a good agreement (88%) between the two methods to provide the same HIV-1 subtype characterization with a good quality score for the NSG run (Q30% = 88%) and with a median total number of reads above 1000 reads. Interestingly, the whole-genome sequencing was more prone to amplify HIV (more fragments) and generate more reads. However, the whole-genome sequencing had a smaller proportion, i.e., 80% versus 95%, of generated reads mapping to HIV-1.

### 3.4. Comparison of NGS and Capillary Electrophoresis (CE) (Sanger) Sequencing

A comparison of the NGS protocol and CE sequencing is summarized in Table 7. To process 24 samples (RT, PR, and INT) by CE sequencing, the time required for waiting, sample preparation, and overall time to result were 81.0 h, 2.0 h, and 83.0 h, respectively. The reagents cost $80 per patient. The full workflow of database processing, analysis, and reporting using DeepChek^®^-HIV was more than 20 min per sample. To process 24 samples (RT, PR, and INT) by NGS, the time required for waiting, sample preparation, and overall time to result were 27.0 h, 4 h, and 31.0 h, respectively. The reagents cost $100–150 per patient. The full workflow of database processing, analysis and reporting using DeepChek^®^-HIV was less than 2 min per sample.

## 4. Discussion

To the best of our knowledge, this is the first study to compare the HIVDR by using a CE-IVD kit for target specifics and whole-genome HIV-1 with validated algorithms (updated version). The NGS method detected all the mutations found by CE sequencing and identified additional mutations of interest.

The global spread of SARS-CoV-2 mobilized both the public and private sector and resulted in a rapid development of solutions focused on SARS-CoV-2 detection and sequencing. Many laboratories are now equipped to perform target sequencing or whole-genome sequencing (WGS). Before the SARS-CoV-2 pandemic, the NGS processing protocol was laborious, expensive, and time-consuming. The total time required for the NGS protocol for the target-specific sequencing was approximately three times longer than the one for the capillary electrophoresis (Sanger) sequencing protocol [23]. In this paper, we presented an improved protocol that reduces the time. The total time to achieve the HIVDR result was 31 h and 83 h for NGS and CE, respectively. Indeed, NGS was faster than CE because all targets (RT/PR/INT or WGS) were pooled and 24, 48, 96, or 384 samples can be multiplexed in one NGS run. The cost of HIVDR using in-house CE (laboratory developed test (LDT)) sequencing was $80/sample [23]. In comparison, the total cost for the pooled NGS was $100–150/sample [23]. The price could be optimized depending on the number of patients per run. To the best of our knowledge, DeepChek^®^ Assay/Software is the cheapest CE-IVD solution [1].

Previous studies have demonstrated that differences between algorithm interpretations do exist with variable degrees of discordances, and using the latest versions is important for patient monitoring [21,25]. The cost of the Abbott Molecular ViroSeq™ HIV-1 solution is >$150/sample, the algorithm interpretations are not up to date, and the solution will be discontinued [8]. Therefore, we need an alternative solution for the laboratories. DeepChek^®^ is an alternative CE or NGS solution, and it allows for an easy selection of different algorithms for a simple interpretation and comparison of the results from different algorithms, while also having well-maintained algorithm versions.

Challenges exist for the standardization and quality assurance of NGS HIVDR genotyping [1]. In this paper, we proposed a standardized CE-IVD solution for the laboratory. Due to the SARS -CoV-2 and variant forms, many laboratories are now equipped to perform whole-genome sequencing (WGS). NGS should occupy a major place in HIV resistance surveillance and clinical care, thanks to its decreasing costs (due to COVID-19 pandemic and the pooling of several application in the same run) and ability to reveal resistant minority variants and study their impact, especially on the new capsid/maturation inhibitors and detection of potential new clinically relevant mutations in the HIV genome.

Several studies showed the importance of detecting minority variants which could not be detected by CE sequencing. Kelentse et al. showed that individuals with HIV-associated cryptococcal meningitis in Botswana harbored minority HIV-1 drug-resistance mutations in RT and protease [26]. Sarinoglu et al. identified a high diversity of protease-site-transmitted drug-resistance mutations in the minority HIV-1variant [27]. El Bouzidi et al. demonstrated that NGS significantly increased the detection of resistance-associated mutations, and the detection of mutation is needed for the newer generation of non-nucleoside reverse-transcriptase inhibitors’ agents [28].

In conclusion, although the costs of reagents for the NGS and CE sequencing are comparable, the NGS protocol is now easier and can be automated to reduce processing times and avoid any mistake that may occur during library preparation to be used in the routine of a clinical diagnostic laboratory. However, combining DeepChek^®^ software with NGS-generated data could allow for better data interpretations in order to ultimately help clinicians provide the most appropriate treatment and improve personalized diagnosis. Indeed, our data demonstrate that this combination allows for an HIVDR status interpretation that is useful for HIV-1 ART monitoring.

## Figures and Tables

**Figure 1 viruses-15-00571-f001:**
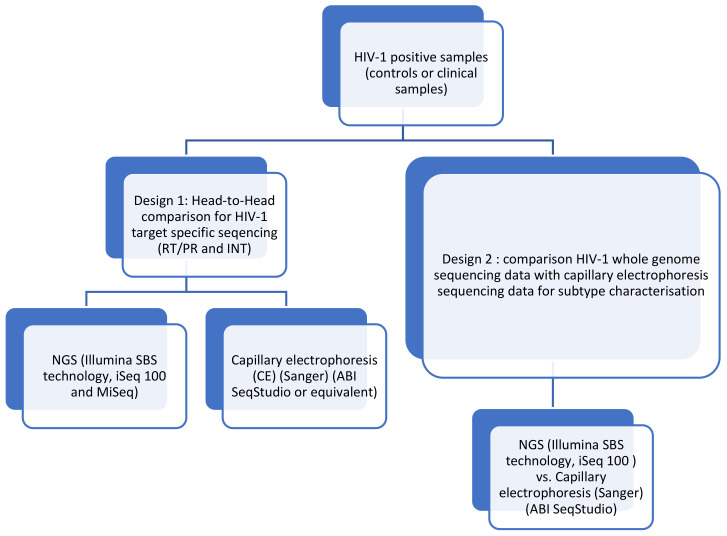
Performance evaluation designs for HIV-1 drug-resistance interpretation using genotyping by sequencing.

**Figure 2 viruses-15-00571-f002:**
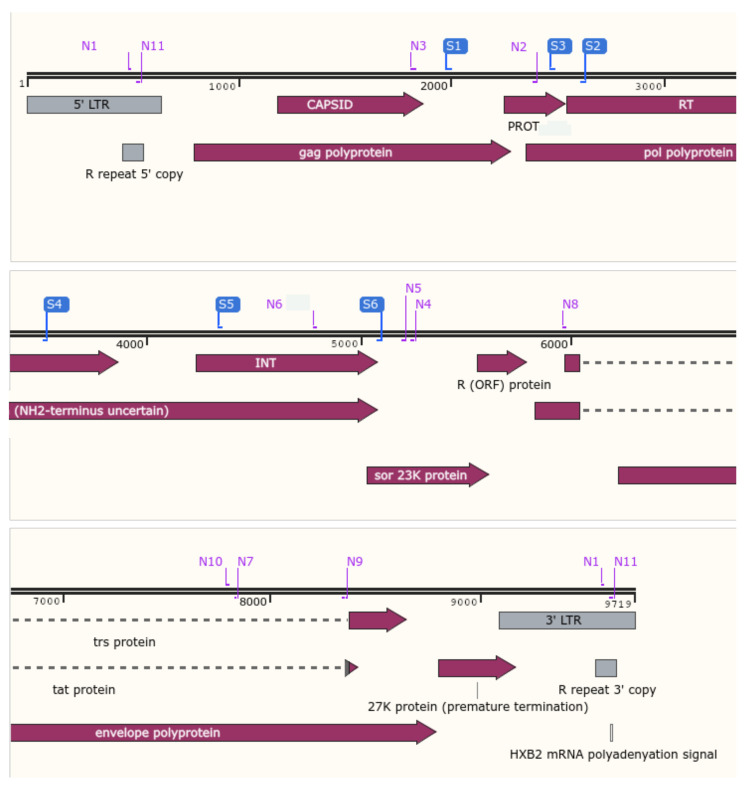
Localization of the CE and NGS primers for HIV drug resistance. In blue, CE primers for reverse-transcriptase, protease, and integrase regions. In purple, NGS primers for the whole-genome HIV (Snapgene Software Version 5.25.5).

**Figure 3 viruses-15-00571-f003:**
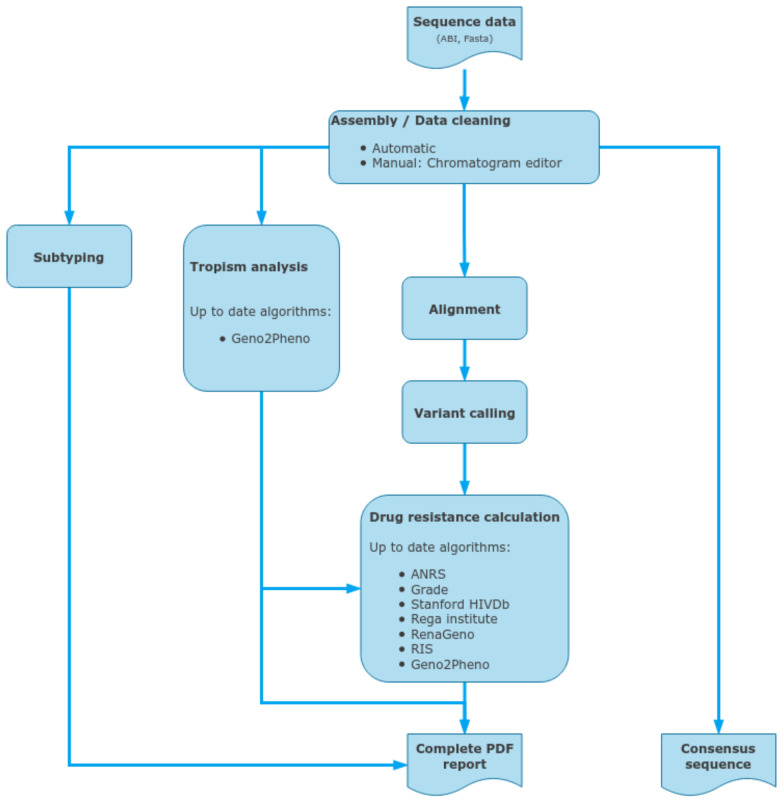
Data inputs and outputs for the ViroScore, a standalone CE-IVD downstream analysis software for clinical reporting, using capillary electrophoresis sequencing outputs.

**Figure 4 viruses-15-00571-f004:**
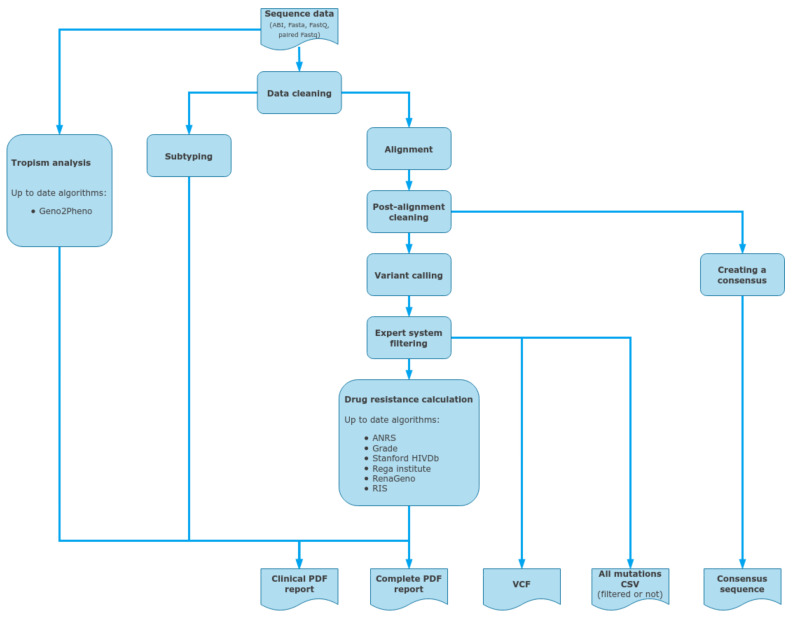
Data inputs and outputs for the DeepChek^®^ software, as a standalone CE-IVD downstream analysis software for clinical reporting for both capillary electrophoresis and next-generation sequencing outputs; such NGS outputs could include HIV-specific targeted genes or HIV whole genome.

**Table 1 viruses-15-00571-t001:** HIV-1 gene associated with antiviral resistance when using the DeepChek^®^ Assay Whole Genome HIV-1 Genotyping.

Anti-HIV Drug Class	HIV-1 Gene Target	Target-Specific Assay (Fragment#)	Whole-Genome Assay (Fragment#)
Capsid inhibitors	gag	-	1
Nucleoside reverse transcriptase inhibitors (NRTIs)	reverse transcriptase	1	2
Non-nucleoside reverse transcriptase inhibitors (NNRTIs)	reverse transcriptase	1	2
Protease inhibitors (PIs)	protease	2	2
Integrase inhibitors (IIs)	integrase	3	2
Integrase strand transfer inhibitors (INSTIs)	integrase	3	2
n.a.	vif, vpr, vpu (accessory proteins)	-	3
Fusion inhibitors	gp41	-	4
Post-attachment inhibitors	gp120	-	4
n.a.	nef (accessory protein)	-	5

**Table 2 viruses-15-00571-t002:** Limit of detection for the HIV-1 target-specific sequencing using iSeq100.

Concentration (cp/mL)	Number of Samples Tested	Number of Correctly Identified Samples	Percentage of Correctly Identified Samples
2000	13	13	100%
1000	10	10	100%
500	10	10	100%

**Table 3 viruses-15-00571-t003:** Optimal NGS median coverage for the HIV-1 target-specific sequencing using iSeq100.

Concentration (cp/mL)	Number of Samples Tested	Samples with Optimal Median Coverage (≥1000)		Samples with Sub-Optimal Median Coverage (>50×–< 1000)	
		Number	%	Number	%
2000	13	13	100%	0	0%
1000	10	10	100%	0	0%
500	10	10	100%	0	0%

**Table 4 viruses-15-00571-t004:** QCMD results for the HIV-1 target-specific sequencing.

		NGS iSeq100 ANRS		NGS iSeq100 Stanford		NGS MiSeq Stanford		CE Stanford			Expected Results
QCMD	Region	Subtype	Mutation of Interest	Subtype	Mutation of Interest	Subtype	Mutation of Interest	Subtype	Mutation of Interest	Subtype	Mutation of Interest
HIVDR 21S_01	RT	C (93,67%)	M41L, E44D, D67N, T69D, A98G, M184V, L210W, T215Y	C (93,59%)	M41L, E44D, D67N, T69D, A98G, M184V, L210W, T215Y	C (93.6%)	M41L, E44D, D67N, T69D, A98G, M184V, L210W, T215Y	C (100%)	M41L, E44D, D67N, T69D, A98G, M184V, L210W, T215Y	C	M41L, E44D, D67N, T69D, A98G, M184V, L210W, T215Y
	PR	C (92,26%)	L10F, G16E, M36V, H69K), L89M	C (92,26%)	L10F, D30N, N88D	C (92.26%)	L10F, D30N, N88D	C (100%)	L10F, D30N, N88D	C	L10F, D30N, N88D
	INT	C (95,71%)	ND	08-BC (95,53%)	ND	08_BC (95.71%)	ND	Not performed	Not performed	C	ND
HIVDR 21S_02	RT	0206 (95,12%)	V179I	0206 (95,12%)	ND	0206 (95.18%)	ND	Unassigned_2;02_AG, A1 (100% similarity)	ND	AG	ND
	PR	02_AG (97,64%)	M36I, H69K, L89M	02 AG (97,64%)	ND	02_AG (97.64%)	ND	02_AG (1) (100% similarity)	ND	AG	ND
	INT	02 AG (97,65%)	ND	02 AG (97,5%)	ND	02_AG (97.65%)	ND	Not performed	Not performed	AG	ND
HIVDR 21S_03	RT	B (99,59%)	ND	B (99,59%)	ND	B (99.58%)	ND	B (100%similarity)	/	B	ND
	PR	B (93,6%)	L10I, L10V, K20R, L33I, M36I, M46I, I54V, L63P, A71T, V82A, L90M	B (93,6%)	K43T (19,63%), M46I, I54V, V82A, L90M	B (93.6%)	K43T (19,69%), M46I, I54V, V82A, L90M	B (96% similarity)	M46I, I54V, V82A, L90M	B	K43T, M46I, I54V, V82A, L90M
	INT	B (99,45%)	ND	B (99,47%)	ND	B (99.45%)	ND	Not performed	Not performed	D	ND
HIVDR 21S_04	RT	D (95,93%)	ND	D (95,93%)	ND	D (95.81%)	ND	D (100% similarity)	ND	D	ND
	PR	D (95,29%)	M36I, D60E, A71T	D (95,29%)	ND	D (95.29%)	ND	D (52% similarity)	ND	D	ND
	INT	D (97,07%)	ND	D (96,97%)	ND	D (97.07%)	ND	Not performed	Not performed	B	ND
HIVDR 21S_05	RT	C (94,49%)	M184V	B (89,2%)	M184V	B (88.81%)	M184V	C (100% similarity)	M184V	C	M184V
	PR	C (93,6%)	G16E K20R, M36I, I54V, H69K, V82A, L89I	B (88,22%)	M46I, I54V, V82A	B (87.88%)	M46I, I54V, V82A	C (100% similarity)	M46I, I54V, V82A	C	M46I, I54V, V82A
	INT	B (91,27%)	ND	B (91,69%)	ND	B (91.27%)	ND	Not performed	Not performed	C	ND

**Table 5 viruses-15-00571-t005:** Aggregated descriptive statistics of external clinical evaluations for the HIV-1 target-specific sequencing using MiSeq.

No. of Samples	301
No. of sites (median number of samples per study)	27 (8)
No. of controls/EQA samples (positive/negative/EQA)	33 (15/6/12)
No. of viral loads available	215
No. of viral loads ≥ 1000 cp/ml	186
Median viral load (cp/mL)	26915
No. of subtypes available	252
% of subtypes B/non-B	63%/37%
No. of PR/RT or of PR/RT/INT DeepChek^®^ Assay ran	91/210
No. of samples with viral load ≥1000 cp/mL and subtype B	149

**Table 6 viruses-15-00571-t006:** Clinical comparisons for the HIV-1 target-specific sequencing.

DeepChek^®^ Assay	Downstream Sequencing Instrument Used with DeepChek^®^ Assay	Device 2 Used for Agreement Concordance	No. of Samples Tested	Concordance (%)
PR/RT + INT	Illumina MiSeq	Abbott^®^ Dx–ViroSeq^®^ HIV-1 Genotyping PR/RT + INT (CE)	23	100%
PR/RT	Illumina MiSeq	LDT (German laboratory) PR/RT (CE)	12	92% *
PR/RT + INT	Illumina MiSeq	Vela Dx–Sentosa^®^ HIV-1 Genotyping PR/RT/INT (NGS)	18	100%

* Only 1 sample was not amplified by DeepChek Assay, but it was not performed a second time. The small number of samples has an impact on the reported figures.

**Table 7 viruses-15-00571-t007:** Comparison of NGS and CE methods for the HIV-1 target-specific sequencing.

Steps	NGS	Time/24 Samples (h)	CE	Time/24 Samples (h)
Sample preparation	RNA extraction kit	1.0	RNA extraction kit	1.0
Amplification	RT-PCR	4	RT-PCR	4
Purification Quantitation	Beads Purification Quality control (TapeStation) Normalization (Qubit)	0.75 0.2 0.5	Enzymatic purification − −	0.2 − −
Library/sequencing reaction	Library preparation	4	Sequencing reaction	2.5
Dilution Sequencing	Dilution and pooling Sequencing	20	Sequencing with SeqStudio 4-capillary	72
Data analysis	FastQ files DeepChek^®^ using ANRS, HIVdb, etc.	0.2	ABI files DeepChek^®^ using ANRS, HIVdb, etc.	1.0
Result	Handling time Waiting time Time to result	4 27 31	Handling time Waiting time Time to result	2 81 83
Price	Reagent cost $/sample	100–150 *	Reagent cost/sample	80
Sensitivity		1 to 3%		20%

* Including extraction, PCR, library preparation, indexes, sequencing, and software.

## Data Availability

Data sharing is not applicable to this article.

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
