# Peer review of "From Capillary Electrophoresis to Deep Sequencing: An Improved HIV-1 Drug Resistance Assessment Solution Using In Vitro Diagnostic (IVD) Assays and Software"

_viruses, 2023, doi:10.3390/v15020571_

Round 1
Reviewer 1 Report
Thank you for proposing this paper, interesting for the search of resistance mutations for HIV. I accept it with minor modifications. It lacks some information and clarification in the core of the paper and needs to review the concordance between the information cited in the abstract and in the results.
INTRODUCTION
The subject of this paper is of interest given the recognised value in the HIV world of being able to detect minority variants for the management of HIV-infected persons. Today, since the arrival of COVID-19, NGS is a method that is increasingly implemented in laboratories. The advantage of this method to search for minority variants for HIV has led some laboratories to switch from the Sanger technique to NGS. The authors could modulate the first sentence of their introduction which gives the impression that only Sanger is currently used. Moreover, the bibliography cited is missing some publications between 2020 and 2022. There is only one publication on Lenacapavir from 2022. The bibliography section needs to be updated.
Specify in the introduction what the targets for HIV genome correspond to.
ABSTRACT
Regarding the summary, it is too detailed (e.g. material and methods section).
Some results are indicated in the abstract but not found in the results obtained (for example the threshold at 106c/ml, the results are carried out on a min viral load of 500c/ml).
Check correspondence between the information cited in the abstract and that to be found in the results.
Lines between 15 and 19 state that two sequencers are used for comparison but not clear in the results.
Does the choice of the 3% threshold apply to all viral loads?
MM
Line 133, the reference of the Sanger sequencer used is missing.
Figure 2, not easily readable.
Table 1, specify the different assays, what do they correspond to?
Missing details on how the two kits are set up.
Paragraph 2.9, how are the results returned, in what format? Is there a section dedicated to quality? Coverage regions, number of reads obtained?
RESULTS
Two kits are evaluated in this paper under different conditions and analysed using Deepchek software. It is sometimes difficult during the reading of the results to understand what was on which sample and on which sequencer. Please specify more precisely what results were obtained and which tests were performed on which sequencer. It would be interesting to refer to what is shown in figure 1 with both designs for more clarity.
Paragraph 3.3, more precision on the results obtained between the comparison of the two kits is expected (are all mutations found in the same sequenced regions?)
For low viral loads, the authors specify two methods (ultracentrifugation without specification and centrifugal filter), only one method is specified in the MM. What is the viral load limit? Is one method more effective than the other?
Paragraph 3.2, to specify which sequencer is used
Line 243, what % of the mutation is present (specify in the text). What are the expected results of QCMD?
Line 247, need more clarification on the sensitivity results specified
.
Author Response
We want to thank the reviewer for his relevant comments.
Thank you for proposing this paper, interesting for the search of resistance mutations for HIV. I accept it with minor modifications. It lacks some information and clarification in the core of the paper and needs to review the concordance between the information cited in the abstract and in the results.
INTRODUCTION
The subject of this paper is of interest given the recognised value in the HIV world of being able to detect minority variants for the management of HIV-infected persons. Today, since the arrival of COVID-19, NGS is a method that is increasingly implemented in laboratories. The advantage of this method to search for minority variants for HIV has led some laboratories to switch from the Sanger technique to NGS. The authors could modulate the first sentence of their introduction which gives the impression that only Sanger is currently used.
The sentence was modified in the abstract (page 1 line 10).
The sentence was modified in the introduction (page 2 line 55).
Moreover, the bibliography cited is missing some publications between 2020 and 2022. There is only one publication on Lenacapavir from 2022. The bibliography section needs to be updated.
3 Publications were added in the introduction: Novitsky et al. 2022, Li et Al.2022 and Armenia et al. 2022
3 Publications were added in the discussion: Kelentse et al. 2022, Sarinoglu et al. 2022 and El Bouzidi et al. 2022
Specify in the introduction what the targets for HIV genome correspond to. The sentence was corrected (page 2 lines 80 and 81)
ABSTRACT
Regarding the summary, it is too detailed (e.g. material and methods section).
We modified the abstract: 380 words (555 words before)
Some results are indicated in the abstract but not found in the results obtained (for example the threshold at 106c/ml, the results are carried out on a min viral load of 500c/ml).
We reached 100% of samples with an optimal median coverage at a concentration of 1000 cp/ml (assay cut-off). Even if we were able amplify adequately HIV-1 subtype B samples with an optimal coverage at a concentration of 500 cp/mL, the intended use of the CEIVD assay was for viral load above 1000 cp/mL and for B subtype).
Check correspondence between the information cited in the abstract and that to be found in the results.
One sentence was deleted because it was not mentioned in the abstract: “On the second set of performance testing, for clinical samples with a median viral load indicative of a failing therapy or untreated patients, 172’736 cp/mL with an interquartile range of (59’912;258’211 cp/mL), the detection rate useful for analysing and interpretating NGS reads was high (> 80%) both for target specific and whole genome.”
Lines between 15 and 19 state that two sequencers are used for comparison but not clear in the results.
Sequencers were clarified (page 4 line 145 and 147)
Does the choice of the 3% threshold apply to all viral loads? Yes, because an ultracentrifugation and systematic nested PCR were performed for low viral load.
MM
Line 133, the reference of the Sanger sequencer used is missing. Reference was added (line 132
Figure 2, not easily readable. Figure 2 has been changed ( page 4, line 125).
Table 1, specify the different assays, what do they correspond to? Missing details on how the two kits are set up. Assays were specified (page 4)
Paragraph 2.9, how are the results returned, in what format? Is there a section dedicated to quality? Coverage regions, number of reads obtained? Format and quality are specified in the Additional information.
RESULTS
Two kits are evaluated in this paper under different conditions and analysed using Deepchek software. It is sometimes difficult during the reading of the results to understand what was on which sample and on which sequencer. Please specify more precisely what results were obtained and which tests were performed on which sequencer. It would be interesting to refer to what is shown in figure 1 with both designs for more clarity. Sequencers were specified. Line 227, 238; page 12 line 2,
Paragraph 3.3, more precision on the results obtained between the comparison of the two kits is expected (are all mutations found in the same sequenced regions?)
2 sentences were added:
Line 242: One hundred percent agreement was found between the iSeq100 and MiSeq system.
Line 245: The mutation K43T on the PR was detected with NGS only for both sequencers, iSeq100 and MiSeq at 19,63% and 19,69% respectively.
For low viral loads, the authors specify two methods (ultracentrifugation without specification and centrifugal filter), only one method is specified in the MM. What is the viral load limit? Is one method more effective than the other? Ultracentrifugation and then centrifugal filter were used for samples with < 1000 copies/mL. This workflow was added (page 3 line 111, 112)
Paragraph 3.2, to specify which sequencer is used Sequencer was added (line 239)
Line 243, what % of the mutation is present (specify in the text). The % was added line 246
What are the expected results of QCMD? Expected results of QCMD were added (table 4)
Line 247, need more clarification on the sensitivity results specified. The sensitivity was specified (Line 250, 252 and 253)
Reviewer 2 Report
This study performed NGS-based WGS on the existing DR mutation assay and will be used as a very useful tool for analyzing genetic diversity of HIV-1 as well as clinical doctors.
1. However, the content of the abstracts is too long and the contents that should be included in the text are written in too much detail, so please shorten them.
2. The most important result of this study is that NGS is used to identify drug-resistant mutations of less than 3% to help clinical treatment, and please emphasize the importance of the results in the discusion.
3. Also, please explain whether it is possible for the genotype of HIV-1 in the discusion and what the results will be based on the genotype through in-silco analysis.
3. Please check the Line 27, "106cp/ml" is correct?
Author Response
We want to thank the reviewer for his relevant comments.
This study performed NGS-based WGS on the existing DR mutation assay and will be used as a very useful tool for analyzing genetic diversity of HIV-1 as well as clinical doctors.
1. However, the content of the abstracts is too long and the contents that should be included in the text are written in too much detail, so please shorten them. We modified the abstract: 380 words (555 words before).
2. The most important result of this study is that NGS is used to identify drug-resistant mutations of less than 3% to help clinical treatment, and please emphasize the importance of the results in the discusion.
This paragraph was added to the manuscript: (page 14 lines: 63-70)
Several studies showed the importance to detect minority variants, and which could not be detected by Sanger sequencing. Kelentse et al showed that individuals with HIV-associated cryptococcal meningitis in Botswana harboured minority HIV-1 drug resistance mutations in RT and protease. Sarinoglu et al. identified high diversity of protease site transmitted drug resistance mutations in the minority HIV-1variant. El Bouzidi et al demonstred that NGS significantly increased the detection of resistance-associated mutations and detection of mutation is needed for newer-generation of non-nucleoside reverse transcriptase inhibitors angents.
Kelentse N, Moyo S, Choga WT, Lechiile K, Leeme TB, Lawrence DS, Kasvosve I, Musonda R, Mosepele M, Harrison TS, Jarvis JN, Gaseitsiwe S. High concordance in plasma and CSF HIV-1 drug resistance mutations despite high cases of CSF viral escape in individuals with HIV-associated cryptococcal meningitis in Botswana. J Antimicrob Chemother. 2022 Dec 23;78(1):180-184. doi: 10.1093/jac/dkac372. PMID: 36322466.
Sarinoglu RC, Sili U, Hasdemir U, Aksu B, Soyletir G, Korten V. Diversity of HIV-1 Subtypes and Transmitted Drug-resistance Mutations Among Minority HIV-1 Variants in a Turkish Cohort. Curr HIV Res. 2022;20(1):54-62. doi: 10.2174/1570162X19666211119111740. PMID: 34802406.
El Bouzidi K, Datir RP, Kwaghe V, Roy S, Frampton D, Breuer J, Ogbanufe O, Murtala-Ibrahim F, Charurat M, Dakum P, Sabin CA, Ndembi N, Gupta RK. Deep sequencing of HIV-1 reveals extensive subtype variation and drug resistance after failure of first-line antiretroviral regimens in Nigeria. J Antimicrob Chemother. 2022 Feb 2;77(2):474-482. doi: 10.1093/jac/dkab385. PMID: 34741609; PMCID: PMC8809188.
- Also, please explain whether it is possible for the genotype of HIV-1 in the discusion and what the results will be based on the genotype through in-silco analysis.
We agree with the reviewer that is important to speak about the in-silico analysis for drug resistance detection but the clinical application for routine lab is complicated because we need validation standards, implementation, and advanced computational requirements. In this study we wanted to show that the implementation of NGS is an easy solution for routine lab with an easy software.
4. Please check the Line 27, "106cp/ml" is correct?the concentration was corrected: 106 cp/ml (page 1 line 27)